# Power Mobility, Supported Standing and Stepping Device Use in the First Two Years of Life: A Case Report of Twins Functioning at GMFCS V

Roslyn W. Livingstone [1,2,*] , Angela J. Chin [1,2] and Ginny S. Paleg [3]

1    Occupational Science and Occupational Therapy, Faculty of Medicine, University of British Columbia, Vancouver, BC V6T 2B5, Canada; angela.chin@alumni.ubc.ca
2    Neuromotor Program, Sunny Hill Health Centre, Vancouver, BC V6H 3N1, Canada
3    Montgomery County Infants and Toddlers Program, Rockville, MD 20825, USA; ginny@paleg.com
*    Correspondence: roslyn.livingstone@ubc.ca

**Abstract:** Mobility experience has a positive impact on activity, participation, socialisation, language and cognition, but children with cerebral palsy (CP), Gross Motor Function Classification System (GMFCS) level V require assistive devices or assistance in all environments. Supported standing devices afford upright, weight-bearing positions to promote muscle, bone, joint and overall health. Supported stepping devices afford stepping and upright independent mobility, positively impacting self-esteem and participation, while power mobility is the only possibility for effective, independent community mobility. These devices and opportunities should be introduced at the age when children who are typically developing are pulling to stand, moving and exploring their environment. A detailed case description including lived experience and device use data is presented for female twins with dystonic tetraplegic CP born at 25 weeks gestational age and functioning at GMFCS level V. The feasibility of using power mobility, standing and stepping devices in home and community settings within the first two years is illustrated. The twins transitioned from spending 24 h in lying positions or being held in arms to spending more than 2 h daily in upright positions and having opportunities to move independently. Positioning and mobility devices can help to address all the F-words for child development: functioning, family, fitness, fun, friends and future.

**Keywords:** powered wheelchair; electric wheelchair; standing frame; stander; gait trainer; cerebral palsy; early intervention; participation; assistive devices

## 1. Introduction

Cerebral palsy (CP) is an umbrella diagnosis that includes all non-degenerative, life-long, sensory-motor impairments. Genetic, anatomical, traumatic, metabolic and other causes are now included [1]. CP can be accurately diagnosed by 3 months adjusted age by combining results of the Prechtl General Movement Assessment (GMA) [2], Hammersmith Infant Neurological Exam (HINE) [3] and neuro-imaging, increasing access to care and early intervention [4].

Children at Gross Motor Function Classification System (GMFCS) IV require wheeled mobility in most settings, while children at GMFCS V have the most limited abilities to maintain position or move independently and require assistive devices or assistance in all environments [5,6]. GMFCS level is more challenging to determine before age 2 and may require reclassification between 2 and 4 years of age [7,8]. Early confirmation of the risk for CP leads to better mental and physical health outcomes for the entire family [9], although parents may need time to adjust to the diagnosis before details of the motor outcome prognosis are shared, and this information may need to be revisited over time [10].

Approximate GMFCS level can now be determined between 3 and 5 months adjusted age using the GMA Motor Optimality Score (GMA-MOS) [11]. A GMA-MOS score below 8

(2–5 months) [12] and a HINE score below 40 (3–24 months) [3] are the current cut-offs for identifying children at highest risk of being GMFCS IV or V. However, these tools appear to be underused in clinical practice [13,14]. Formal parent training and assistive devices are the most effective interventions for very young children at GMFCS IV or V [15], and since assistive devices take time to procure, knowing that the child is at high risk to be non-ambulant allows early initiation of this process.

Motor type and severity are more challenging to determine in infancy, as motor disorders such as spasticity and dystonia develop and change over the first two years [16]. Children with features of both spasticity (velocity-dependent resistance to stretch) and dystonia may be described as having dyskinetic or mixed CP [17]. Dystonia has been described as fluctuating hypertonia, involuntary movement and postures, and may be caused by overflow from intentional movements [18]. The Hypertonia Assessment Tool (HAT) [19] distinguishes dystonia from spasticity in children with CP and may be used in combination with the Barry-Albright Dystonia Scale (BADS) [20] to determine the presence and severity of dystonia [21]. Reliability of the HAT is greater for spasticity [19,22], possibly due to variability in dystonia. Dystonia is frequently seen in combination with spasticity [19,22,23], and children with both dystonia and spasticity may have more severe functional limitations than children with spasticity alone [24]. Dystonia is associated with reduced movement and increased tone, [23] and severity increases with GMFCS level [21].

Prolonged static positioning in children with limited abilities to change position leads to pain and fixed postural asymmetries [25]. Provided from 3 months adjusted age, adaptive seating affords practice of grasp and reach, and positively influences vision, communication and social interaction [26]. Supported standing in a device can begin as early as 9 months adjusted age, and positively influences muscle, joint and bone health as well as enhancing functioning and participation with others [27]. Supported stepping is feasible from 9–15 months and can improve head control, increase access to the environment and positively influence a sense of autonomy and self-esteem [28]. Power mobility with extensive adaptations is the only means of efficient, independent functional mobility for children functioning at GMFCS V [5,6,29,30] and may be introduced around 12 months [29].

Self-initiated independent movement in early childhood and opportunities for exploratory behaviour are thought to promote a developmental cascade, positively influencing later abilities. Infants who were able to explore toys and their environment more actively by 5 months of age were shown to have greater academic abilities by age 14, independent of other factors [31]. Children who are more limited in their self-initiated exploration appear to have greater difficulties with spatial cognition (especially mental rotation) than others with similar learning abilities [32]. ON Time mobility (including upright positioning and movement through space by 9–12 months) is a human right [33], and children at GMFCS V also require opportunities to be upright and engage in independent exploration at the same age as their peers. Clinicians need to ensure that they are addressing the components of functioning, family, fitness, fun, friends, and future in their interventions with all young children with CP, including those at GMFCS V.

These F-words for child development [34,35] are a family-friendly adaptation of the International Classification of Functioning, Disability and Health (ICF) [36]. Functioning relates to the ICF concepts of activity and participation; family is the primary environmental contextual factor for young children; and fitness relates to body structures and function. Fun relates to personal factors and participation, while friends also combines participation and personal factors. Future is not explicitly included in the ICF but encompasses expectations and dreams. The application of the F-words to early intervention for children with non-ambulant CP has been further developed by De Campos and colleagues [15]. Interventions promoting family goals, coaching, routines-based interventions, parent/caregiver training and support and family-centred care may be considered part of the F-word family. Future includes interventions designed to help prevent known longer-term complications of CP (e.g., hip displacement, contractures, etc.) as well as interventions designed to help children,

families, communities and society 'see' children with non-ambulant CP differently, thus promoting future health and opportunities.

Supported standing increases visual and physical access to activities (functioning), affords standing to participate with others (family), promotes bone mineral density and cardio-vascular function (fitness), enables standing to engage in enjoyable activities (fun), promotes being face-to-face with others (friends), enhances hip health and helps prevent contractures (future) [27]. Supported stepping increases head and arm/hand control (functioning), affords participation in family activities and addresses parent goals (family), promotes active movement and energy expenditure (fitness), enhances active play with others (fun), affords face-to-face interaction with others (friends), and enhances self-esteem, autonomy and physical health (future) [28]. Power mobility experiences promote independent mobility (functioning), participation in family outings and activities (family), improves the sleep–wake cycle (fitness), affords enjoyment of movement for its own sake (fun), enhances ability to play with others (friends), and enhances sense of autonomy, independence, cognitive and language development (future) [37].

The purpose of this manuscript is to describe the feasibility and use of adaptive seating, supported standing, stepping and power mobility devices by twins functioning at GMFCS V within the first two years of life (to 24 months chronological age). This case report includes quantitative data and a reflection on how the introduction and use of these devices assisted in addressing the F-words for child development includes lived-experience (qualitative) data.

## 2. Materials and Methods

Case details are taken from a retrospective chart review with parent consent and child assent. Details were confirmed with parents and through the review of videos, either provided by the family or taken by the first author for educational purposes. At time of writing, children are 11 years old and capable of understanding assent for research participation and publication. Assent was communicated through verbal word approximations and forms signed with hand-over-hand assistance. Children were very enthusiastic to share their experiences with others. See Table 1 for details of measures and classifications used in this case report.

**Table 1.** Measures and classifications.

| Measure | Description |
| --- | --- |
| Gross Motor Function Classification System (GMFCS) [5,6] | Five-point ordinal scale classifying gross motor and mobility function in children and youth with CP. Abilities range from I (walks without restrictions in the community by school-age to V (requires support to sit and primarily uses wheelchair mobility) |
| Mini Manual Abilities Classification System (mini-MACS) [38] | Five-point ordinal scale classifying manual abilities in children with CP under 4 years of age. Manual abilities range from I (handles objects easily and successfully) to V (does not handle objects) |
| Communication Function Classification System (CFCS) [39] | Five-point ordinal scale classifying abilities of children with disabilities to send and receive communication. Communication abilities range from I (sends and receives with familiar and unfamiliar partners effectively and efficiently) to V (seldom effectively sends or receives, even with familiar partners) |
| Visual Function Classification System (VFCS) [40] | Five-point ordinal scale classifying how toddlers, children and youth with CP use visual function in daily life. Visual function ranges from I (uses visual function easily and successfully in vision-related activities) to V (does not use visual function even in very adapted environments) |
| Mini Eating and Drinking Abilities Classification System (mini-EDACS) [41] | Five-point ordinal scale classifying eating and drinking functional abilities in children with CP under 36 months of age. Eating and drinking abilities range from I (eats and drinks safely and efficiently) to V (unable to eat or drink safely—tube feeding may be considered to provide nutrition) |

**Table 1.** *Cont.*

| Measure | Description |
|---|---|
| Level of Sitting Scale (LSS) [42] | Eight-point ordinal scale classifying postural abilities of children with disabilities to maintain bench sitting with feet unsupported. Abilities range from 1 (unable to be supported in upright sitting by one adult for 30 s) to 8 (able to move in and out of the seated position in all directions) |
| Posture and Postural Abilities Scale (PPAS) [43,44] | Seven-point ordinal scale classifying postural abilities plus a detailed description of posture from the front and the side in supine and prone lying, bench sitting with feet supported and standing positions. Postural abilities range from 1 (unable to maintain position without support) to 7 (able to move in and out of position). The PPAS is valid and reliable for use with children and adults with CP and can be used as an outcome measure to record change in posture following provision of adaptive seating, lying or standing devices. |
| Barry-Albright Dystonia Scale (BADS) [20] | Five-point criterion-based, ordinal scale from 0 (no dystonia) to 4 (severe dystonia) in eyes, mouth, neck, trunk and limbs for a maximum score of 32 points. BADS is considered to have moderate construct validity, limited content validity and moderate concurrent or predictive validity [45]. |
| Assessment of Learning Powered mobility use (ALP-tool 2.0) [46] | Eight-point ordinal scale describing the power mobility learning process from 1 (novice) to 8 (Expert). It is validated for all ages and cognitive levels. Very good inter-rater reliability has been established between professionals and family members or caregivers [47] |
| Power Mobility Training Tool (PMTT) [48] | Five-point ordinal scale describing non-motor, motor and driving skills from 0 (not attempted) to 4 (able to complete independently >90% of the time) for a total score of 48. It was developed to guide power mobility training for young children. |
| Universal Assessment of Learning Process (Universal ALP-tool) [49] | Modified from the ALP-tool 2.0 to describe the learning process for all tools and assistive devices. For devices activated through physical or body movements such as a stepping device: in stage 1 (phases 1–3), individuals can exert force and learn what the device is used for; in stage 2 (phases 4 and 5), they can grade force and begin searching to find a working pattern for functional use; and at stage 3 (phases 6–8), they can direct force and begin to use the device to attain a functional outcome or achieve a self-selected goal. Phase 3 represents basic use of tool functions, phase 6 is competent use, and phase 8 represents expert tool use integrated into everyday life. |

Other than GMFCS and LSS, remaining measures were either not yet developed and available during the timeline of this case report (2013–2014) or were not standard practice in this clinical setting at that time. Measures or classifications not completed or fully documented at the time were completed retrospectively from the medical record, and video review, by a consensus of a team of researchers: one conducted assessment and intervention during the time-line of the report (RWL); one is the twins' current positioning and mobility therapist and previous school therapist (AJC); and one has extensive experience in early identification and early intervention for children with non-ambulant CP (GSP).

A semi-structured interview took place at the family home with both parents, the twins and their older sister. The interview was conducted by the second author (AJC) and explored how family members recalled their use of assistive devices in early childhood, their views regarding benefits and how they thought this impacted the twins' lives and opportunities. The interview lasted one hour and was audio-recorded with informed consent of all participants. The ethics review for publication of this case report was waived by University of British Columbia Children's and Women's Health Centre of British Columbia Research Ethics Boards.

The interview was transcribed verbatim by the second author, and all three authors used the F-words framework to independently code the transcript. Codes were agreed through consensus and discussion, and quotes selected to illustrate each F-word. In addition, the entire transcript was downloaded into a free word cloud app (www.classic. wordclouds.com). A word cloud represents the frequency/importance of different words, concepts or ideas by relative size. For the purposes of this article, incidental words (e.g.,

conjunctions) were deleted, and words that were part of a concept, phrase or idea (e.g., 'doing things together') combined. This allowed a figure to be created, summarizing the entire interview, illustrating the family's priorities, strengths and core values.

### 3. Description of Feasibility and Use of Assistive Devices in the First 2 Years

Jayde and Skyla (not pseudonyms since children and parents strongly preferred to use their own names) are non-identical female twins born at 25 weeks' gestation. Jayde experienced Grade III and Skyla Grade IV intraventricular haemorrhage during their time in the neonatal intensive care unit (NICU) where they spent over 4 months prior to discharge home to their community. The girls (as they are referred to in their family) are the youngest of four children and have an older sister and half-brother. The girls were seen within the first month of NICU discharge by an infant development consultant who immediately involved a highly experienced early intervention physiotherapist (PT). Within a few months, an occupational therapist (OT) became involved to assist with positioning, feeding, play and other activities of daily living. PT and OT introduced upright supportive seating options from 3–5 months adjusted age including a commercial high-back contoured foam seat on a floor base, and an adaptive seating insert with trunk laterals. However, these options were not successful, and they requested referral to specialized seating and mobility services by 8 months adjusted age.

An overview of both girls' function according to a range of classifications by chronological age of 2 years is provided in Table 2. Although there are no functional classification differences between them, the girls have different personalities, likes and dislikes, and there are subtle differences in their motor control. Skyla has better head and upper limb control, and her dystonia is less severe. However, Jayde has a more outgoing and adventurous nature, and was always willing to try new things first, while Skyla watched to see what happened.

**Table 2.** Functional profile of the twins at 24 months of age.

| Classification | Functional Level at 2 Years of Age |
|---|---|
| Gross Motor Function Classification System (GMFCS) | **V** (has difficulty controlling head and trunk posture in most positions; uses adaptive seating to maintain position comfortably; usually lifted by another person to move about indoors) |
| Mini Manual Abilities Classification System (mini-MACS) | **V** (does not handle objects; can touch, press, hold onto or handle a few objects with constant adult assistance) |
| Communication Function Classification System (CFCS) | **IV** (inconsistent sender and receiver with familiar communication partners) |
| Visual Function Classification System (VFCS) | **I** (uses visual function easily and successfully in vision-related activities) |
| Mini Eating and Drinking Abilities Classification System (mini-EDACS) | **IV** (eats and drinks with close attention to food texture, fluid consistency and the way in which food is offered; supplementary gastrostomy tube feeds were considered and added by age 3) |
| Level of Sitting Scale (LSS) | **2** (requires support from the head down when placed in bench sitting with feet unsupported) |

### 3.1. Initial Assessment

At 9 months adjusted age, the girls had significant fluctuating tone or dystonia affecting the whole body. With or without head rotation, one arm was typically held in a high guard position and the other in extension. This patterning could happen to either side, although Skyla tended to flex the arm on the right side, while Jayde tended to flex on the left. Hip and knee extension with back arching were common, although hip flexion and abduction could be seen on the same side as the flexed upper limb in supine or reclined positioning. At times, dorsi-flexed ankles with extended toes were seen with hip and knee extension. Their supine lying posture is illustrated in Figure 1. On the PPAS chart, a score of 0 indicates that the body segment was neither straight nor aligned. A score of 1 indicates straight and aligned, whereas 0/1 indicates that body segment position was variable.

**Figure 1.** Asymmetrical supine posture at 9 months adjusted age.

Flexing the hips and knees helped relax their tone but resulted in the trunk and neck collapsing into flexion. The girls used baby bouncy seats and reclined positioning in their side-by-side twin stroller, but upright positioning was only possible for very short periods when held in a familiar adult's arms. Total body extension with back arching and neck extension was frequently seen with stimulation or attempts to move.

Although formal measures of dystonia were not completed at the time, from retrospective chart review and family videos, the girls would have scored in the severe range for neck and limbs, moderate (Skyla) to severe (Jayde) in the trunk, moderate for mouth (causing feeding difficulties), and slight for eyes on the BADS. The HAT was developed for children 4 years and older [19] and is not suitable for infants due to the requirements to follow directions [16]. A spastic catch and slight resistance to passive range of motion was noted in the hamstrings, hip adductors and gastrocnemius during hip surveillance clinic around 14 months adjusted age.

Initially, the family goals were to find suitable seating for feeding and play, as it was challenging to hold the girls and assist them to engage in these activities. Parents were also open to trying any equipment that would give the girls new experiences of position and mobility and maximize their participation in family life and activities. Goal-setting was completed with the infant development consultant as part of a family-centred care plan.

Contoured foam seats were soft and did not have any hardware where the girls could hurt their arms or legs, but they did not provide sufficient lateral trunk or head control. The twins would often twist sideways, and the chest straps would cut into their necks. Adaptive seating inserts with trunk and pelvic laterals were too rigid, and the girls would go into full body extension when parents attempted to put them into this type of seat. The girls were level 1 on the PPAS for both sitting and supine positions since they were unable to maintain symmetrical positions without support.

### 3.2. Equipment Trials and Introduction of Positioning and Mobility Devices

Table 3 provides a summary of the time-line of postural and mobility device use between 5 months (adjusted age) and 24 months of age (chronological).

We had initial success with modifying the highchair by adding a small 'wrap around' seat that provided soft circumferential trunk support and fitted inside the commercial highchair. This still did not provide adequate head support, but it was a safe way to begin more upright positioning. Since the twins were at high risk for hip subluxation due to their GMFCS level [50], limited abilities to change position [25] and increased tendency to assume asymmetrical postures [51–53], we introduced abducted hip positioning [54].

**Table 3.** Timeline of device assessment and introduction.

| | 5–8 Months | 9–11 Months | 12–14 Months | 15–18 Months | 19–24 Months |
|---|---|---|---|---|---|
| Adaptive Seating | Commercial contoured foam seat | Custom wrap around seat in highchair | Custom saddle seat with tray on high-low base for feeding and play at home | | |
| Supported Standing | | Equipment trials | Loan program—supine standing frame | | |
| Supported Stepping | | Equipment trials | Anterior support stepping device with custom foam tray | | |
| Power Mobility | | | Specialized early power mobility device | Switch-adapted ride-on toys outdoors at home | Mini power wheelchair in therapy sessions |
| Manual Mobility | Standard twin stroller—reclined positioning | | | Loan program—dynamic manual wheelchair with custom adaptive seating | |

A custom bolster (i.e., saddle) seat with lateral trunk supports, lateral pelvic support, anterior trunk support, anterior shoulder support, head support and large padded tray positioned at mid-trunk height was created for eating and play, on a high-low base. This promoted upright head and trunk positioning, while allowing the hips and knees to assume varying degrees of flexion or extension as the girls' tone fluctuated. Figure 2 illustrates change in PPAS scores when using the saddle seat in comparison with the commercial contoured foam seat. Although trunk laterals were able to stabilize the trunk to the level of the axilla, the upper trunk and shoulder position was influenced by arm and head position and varied.

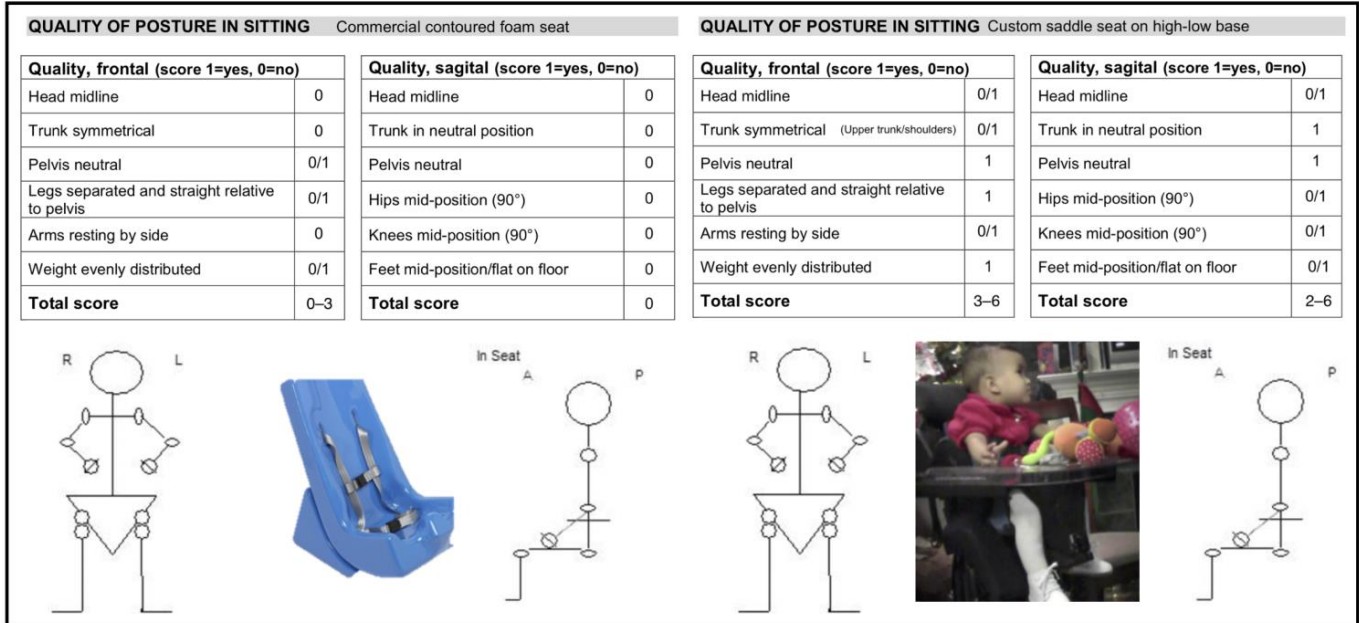

**Figure 2.** Sitting posture comparison between commercial foam and custom saddle seats.

Around 12 months adjusted age, a specialized early power mobility device that could be used in sitting or standing positions was loaned for use at home. It was modified with a large foam bolster and set up in a slightly prone and semi-standing position. A single switch allowed the experience of self-controlled mobility in either a forward or turning direction. Over time, the girls understood that the switch made the device move, even if they needed help to activate or release it.

Around the same time, the girls were trialling various stepping devices with their PT. A forward leaning position with full anterior trunk support was needed for initiation of stepping. They were unable to maintain head control without their arms being supported at mid-chest level on a large soft surface. Funding was requested through medical insurance for an anterior stepping device with a seat, contoured chest support and a large custom padded tray.

Figure 3 illustrates posture in the supine stander. Once the girls were fitted for ankle foot orthoses (AFO's), upright standing and weight-bearing were introduced. A recycled supine stander with a large tray was available on loan with pelvic and trunk laterals, headrest, chest harness, abduction block, individual knee straps and shoe positioners. It was used in the home daily, starting at 15 min each and increasing to 30–40 min each daily. The girls required the same positioning supports and were able to share this stander without components needing to be adjusted.

**QUALITY OF POSTURE IN STANDING**   Custom supine standing frame with tray

| Quality, frontal (score 1=yes, 0=no) | | Quality, sagital (score 1=yes, 0=no) | |
|---|---|---|---|
| Head midline | 0/1 | Head midline | 0/1 |
| Trunk symmetrical   (Upper trunk/shoulders) | 0/1 | Trunk in neutral position | 1 |
| Pelvis neutral | 1 | Pelvis neutral | 1 |
| Legs separated and straight relative to pelvis | 1 | Legs straight, hips & knees extended | 1 |
| Arms resting by side | 0 | Feet mid-position/flat on floor | 1 |
| Weight evenly distributed (through both feet) | 1 | Weight evenly distributed (through the feet) | 1 |
| **Total score** | 3–5 | **Total score** | 5–6 |

**Figure 3.** Posture in supine standing frame.

Funding was approved, and the stepping device was delivered around 13 months adjusted age. Skyla was able to take reciprocal steps with bare feet and could move several feet at a time down the hallway or across the living room. On the Universal Assessment of Learning Process (ALP), she achieved phase 3, being able to exert force consistently and move in a forward direction but could not yet sequence movements in order to grade speed, steer or turn. Jayde spent more time standing but could initiate steps at times and mainly achieved ALP phase 2, being able to exert force intermittently. They were both happy to be in the device for 20–30 min daily at home, enjoying the opportunity to be active and upright for play with parents or siblings.

Figure 4 illustrates upright positioning and mobility experiences.

Around 15 months of age, the girls needed opportunities to play outside, and a switch-adapted ride-on-toy car was loaned. A high-back contoured foam seating insert and soft foam collar were used to provide postural support, and a 4 inch switch was mounted using an adjustable arm. The family was able to use this without therapist support once set up with the equipment—and the twins could have turns to play with their siblings outside on paved areas or on grass at the park.

By 18 months adjusted age, the girls needed opportunities to learn to do more than go and stop. A mini powered wheelchair was loaned to the child development centre for multiple children to use for training. Although the girls had most control over head movements, at this age parents preferred to encourage use of hands for switches and trying a joystick with modified handles and splints. The girls demonstrated understanding of cause–effect but required assistance to use more than one switch or to steer in different directions with a joystick.

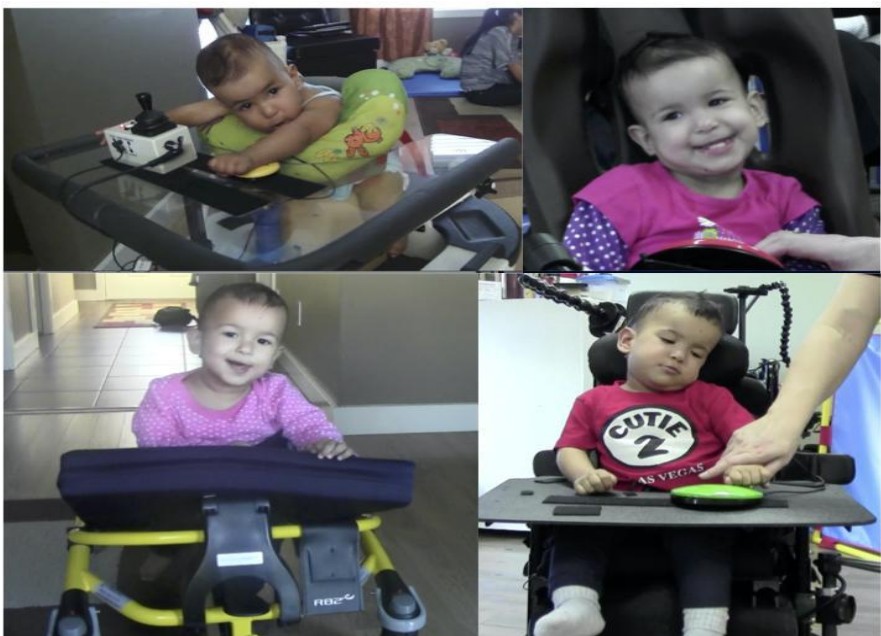

**Figure 4.** Upright mobility experiences. Clockwise from top left; power mobility device in the home, used with single switch access; switch adapted ride-on toy car for outdoor play; exploring power mobility access options in therapy sessions; independent exploration at home in anterior stepping device with large padded tray for postural support.

Around 18 months adjusted age, parents were frustrated that the girls were either supine in the stroller and unable to participate in family outings, or uncomfortable and crying when positioned more upright. Two dynamic manual wheelchairs with custom adaptive seating became available through the provincial loan program and were introduced for community mobility. Figure 5 illustrates typical activities for the girls around chronological age 2 years, complete with participation in family routines incorporating multiple assistive devices.

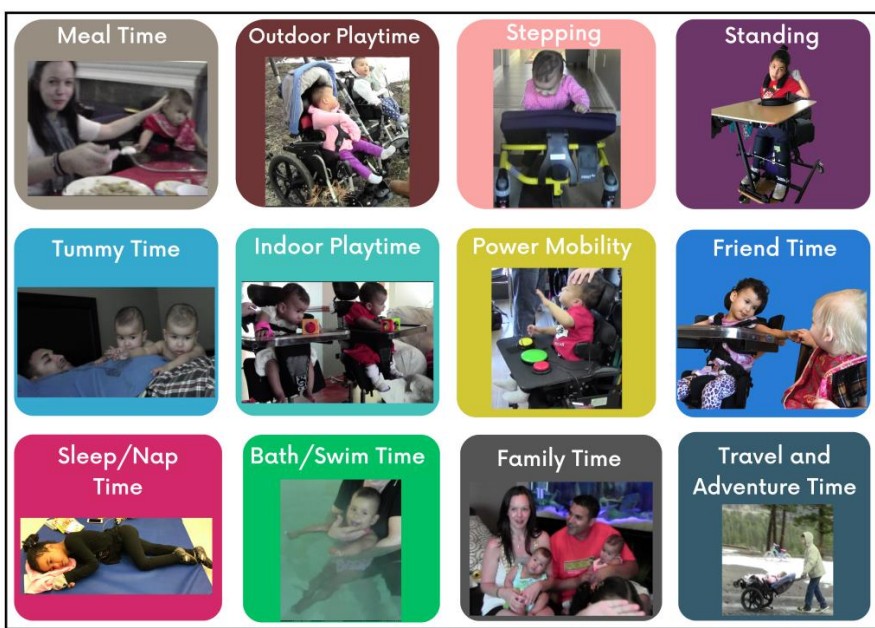

**Figure 5.** Routines-based use of positioning and mobility devices.

Table 4 illustrates the progression of power mobility skills in the different types of devices—from the specialized early power mobility device at 12 months, the ride-on toy car at 15 months, and the mini power wheelchair during therapy sessions at 18 and 24 months.

**Table 4.** Power mobility skill progression from 12 months (adjusted age)–24 months (chronological).

| | Jayde | | | | Skyla | | | |
|---|---|---|---|---|---|---|---|---|
| Age in Months | 12 | 15 | 18 | 24 | 12 | 15 | 18 | 24 |
| ALP | 1 | 2 | 3 | 3 | 1 | 2 | 3 | 3 |
| PMTT Non-Motor Skills | | | | | | | | |
| Cause–effect: movement | 1 | 2 | 3 | 4 | 1 | 2 | 3 | 4 |
| Cause–effect: direction | 0 | 0 | 1 | 2 | 0 | 0 | 1 | 2 |
| Stop and go | 1 | 2 | 3 | 3 | 1 | 2 | 3 | 3 |
| Visual skills | 3 | 4 | 4 | 4 | 3 | 4 | 4 | 4 |
| PMTT Motor Skills | | | | | | | | |
| Activate | 1 | 1 | 1 | 2 | 1 | 1 | 2 | 2 |
| Release | 1 | 1 | 1 | 1 | 1 | 1 | 1 | 2 |
| Sustain > 5 s | 1 | 1 | 1 | 1 | 1 | 1 | 2 | 2 |
| PMTT Driving Skills | | | | | | | | |
| Forward 5 feet | 0 | 1 | 1 | 1 | 0 | 1 | 1 | 1 |
| Turn right | 0 | 0 | 1 | 1 | 0 | 0 | 1 | 1 |
| Turn left | 1 | 0 | 1 | 1 | 1 | 0 | 1 | 1 |
| Reverse | 0 | 0 | 0 | 0 | 0 | 0 | 0 | 0 |
| Maneuver | 0 | 0 | 0 | 1 | 0 | 0 | 0 | 1 |
| PMTT Total | 9/48 | 12/48 | 17/48 | 21/48 | 9/48 | 12/48 | 19/48 | 23/48 |

ALP: Assessment of Learning Powered mobility tool version 2.0; PMTT: Power Mobility Training Tool.

### 3.3. Family's Lived Experience Reflected in the F-Words

The girls were seen at home with their parents and older sister to reflect on their early use of assistive devices. The word cloud (Figure 6) is based on the entire interview and illustrates family priorities, strengths and core values prominent in the discussion.

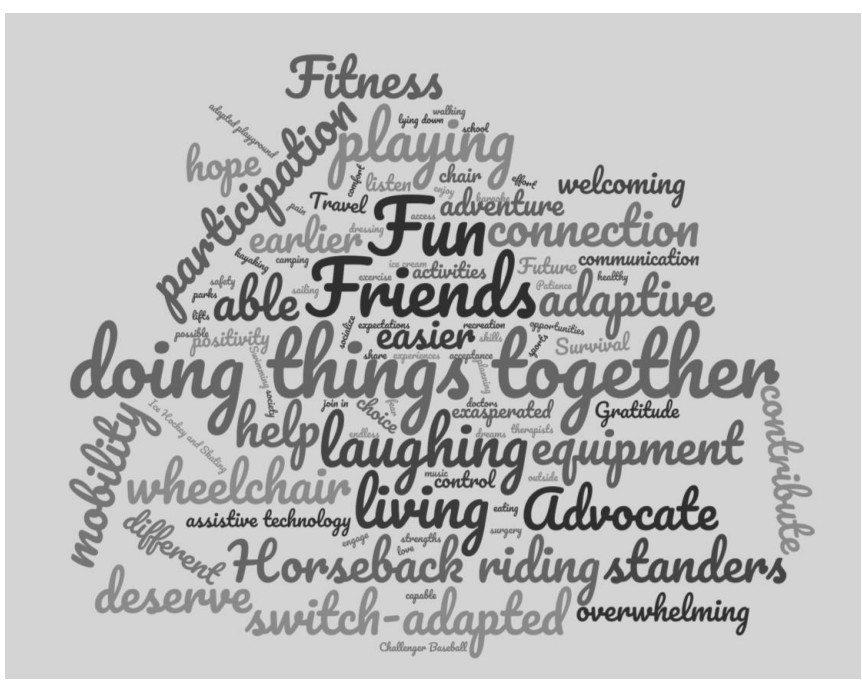

**Figure 6.** Family priorities, strengths and core values.

The F-words (functioning, family, fitness, fun, friends, future) are a family-friendly adaptation of the ICF and designed to facilitate knowledge translation. We used the F-words (highlighted with bold text) to structure the reporting of themes that emerged during the interview. Themes are written in question form and illustrated by quotes from the family.

- How did adaptive equipment help with the girls' early development and **functioning**?

  **Mom:** *I think seeing the world not laying down. . .as soon as you got them up in the feeding chairs, that allowed full participation in eating. . .communicating, and learning how to socialize, and other equipment. . .I guess it's more for **fun**, playing like typical [kids]. . . to be experimental with speed and direction. . .like the GoBot, spinning in a circle. . .or going in the mini-car and being able to control that.*

- How did adaptive equipment help the girls engage and participate in family life and routines?

  **Mom:** *Like in every aspect . . . like being able to move, being able to go out, and being able to eat safely, and being able to play, and engage, and socialize. And experience life. Everything!*

- How has adaptive equipment helped the girls with **fitness**?

  **Sister:** *There's standing frames*

  **Dad:** *And keeping the body in proper form, right? Like spines, hips, arms, legs, neck, everything right? Like it's been huge.*

- Does your equipment help you with making **friends**?

  **Mom:** *you had your standing frame at school and you participated differently with gym. You have had some fun with your power wheelchair with your friends.*

  **Dad:** *I think kids think it's cool. . . their power wheelchair where it can go as almost as high as me, which is six feet. Yeah, it's cool. I think it brings in a crowd, especially if it's out in public.*

  **Mom:** *And having assistive technology for eye gaze and stuff. Because other kids have their laptops at school now and they have their own systems. That's really important.*

- How has having this equipment impacted **family** life and the girls' **future**?

  **Dad:** [without it] *We wouldn't be able to go anywhere, right? . . .The pros far outweigh the cons. . . it's definitely worth it, because without it we'd just be, honestly, I don't know. I can't fathom life without it.*

  **Sister:** *The girls have more opportunities*

  **Mom:** *So, if you look back at ten years of say, them laying on the couch without support, or laying on the ground, or not being able to sit upright, or . . . experience different fun . . . equipment, along with friends and peers, and being able to be out in the world—well, then you've got two completely different kids right now. They would look a lot different. They would act a lot different. They wouldn't have any speech or quality of life.*

  **Dad:** *you can see how much it has affected their lives and how much it's made our lives better.*

- What advice would you give to other families in a similar situation?

  **Mom:** *Try not to compare your children to others around you . . .Also to keep on fighting for what your child and you deserve.*

  **Dad:** *Advocate. . .Stay positive. . . because there's always something around the bend that you're gonna have to deal with. Right? So, staying positive, Number one. But number two, surround yourself with good support—and that means friends, family, therapists. . .*

  **Mom:** *it was really overwhelming and stuff, right? But. . .what would it be like. . .if we didn't start earlier? I believed in everybody around us, I guess. And then just to try and live every day the best you can. Because you're going to have lots of bad days.*

Functional classifications remain consistent for both girls from 2 to 11 years of age, and they continue to enjoy tastes, although nutritional needs have been met via gastrostomy tube since 3 years of age. By age 5, hip subluxation progressed on the more commonly extended hip for both girls (right for Jayde and left for Skyla); it was successfully treated with unilateral varus de-rotation osteotomy surgeries, and hips remain pain-free. Hip and spine surveillance are ongoing, but there are no concerns or contractures. Unfortunately, over the last few years, Jayde developed more severe dystonia and has related pain and medical complications at this time. Skyla can use her power wheelchair with stand-by supervision in open spaces, and also uses a hands-free stepping device. Both girls continue to use supine standers and successfully use eye gaze-enabled technology for schoolwork and communication. Figure 7 illustrates the girls' current F-words profile.

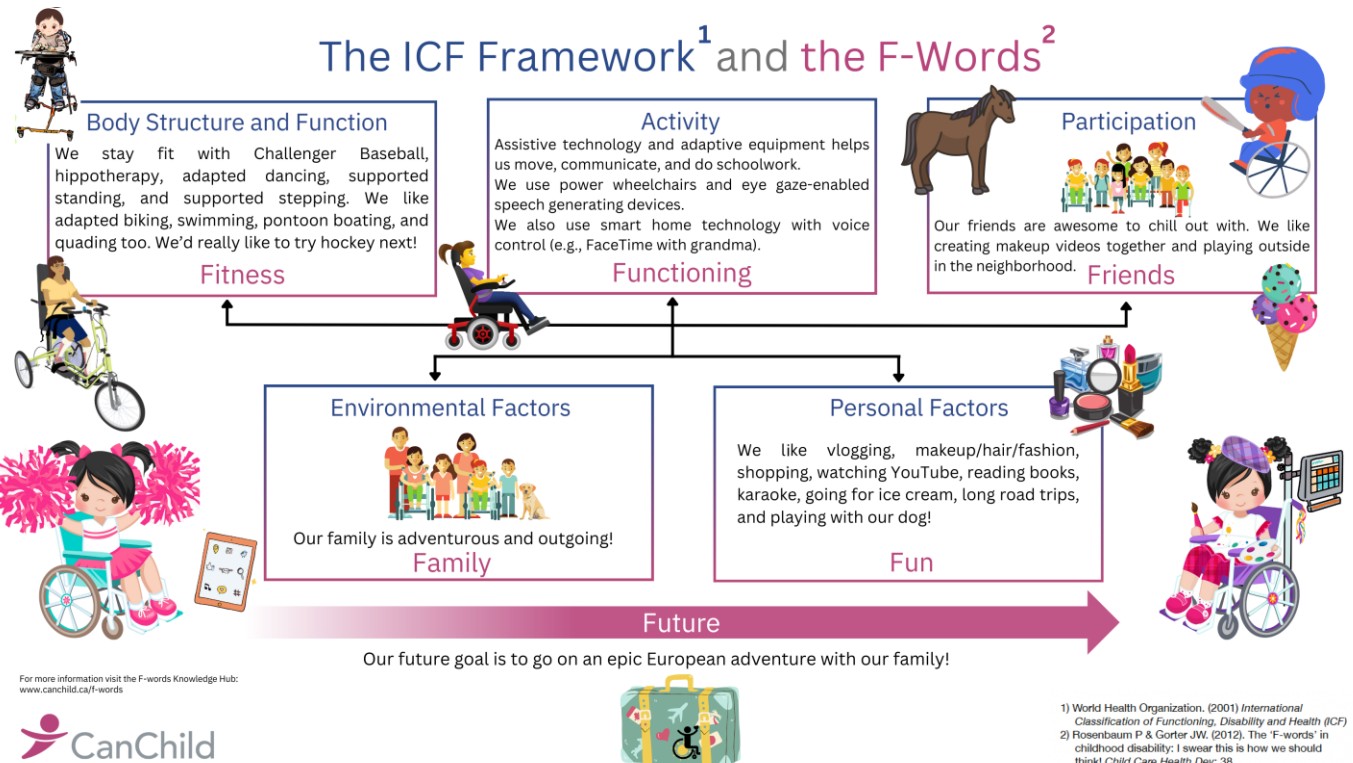

**Figure 7.** F-words profile at 11 years of age [34,36]. The graphic is adapted with permission from CanChild's original, and subsequently modified, integrated ICF and F-words poster. CanChild's F-words Knowledge Hub: www.canchild.ca/f-words. Copyright 2023, McMaster University.

## 4. Discussion

This case report highlights the importance of early diagnosis and identification of motor prognosis and movement disorder, in order to initiate appropriate early intervention. Introduction of assistive devices in the first two years was feasible for the family in this case report, and a positive influence on all the F-words for child development (functioning, family, fitness, fun, friends and future) is illustrated in their lived experience. For all GMFCS levels, early interventions should be goal-directed, child-active, caregiver-delivered, based in natural environments and routines and allow practice and problem solving [55]. For children anticipated to function as GMFCS IV or V, environmental enrichment includes the ON-Time (age-appropriate) introduction and use of assistive devices [31], and postural management to promote functioning and participation as well as to help promote future health and prevent pain associated with contractures and fixed asymmetries [54].

Although formal measures such as the GMA and HINE were not standard practice at the time, experienced early intervention practitioners (infant development consultant, PT and OT) recognized the complexity of the twins' presentation and supported parents in

initiating referrals to specialty services and trialling appropriate assistive devices. Parents were supported in using these devices at home to maximize the girls' participation in family life. This is in line with recent recommendations that the most effective intervention strategies for infants anticipated to function as GMFCS IV or V are formal parent training and assistive devices [15].

Static seating and positioning interventions were not effective due to the twins' severe dystonia, as they did not accommodate their severe extensor tone and asymmetrical patterns. Dynamic seating is frequently recommended for individuals with dystonia to reduce injury and pain, and to increase stability and function [56–60]. Since dynamic seating options are not available for infants, a custom saddle seat was created, prioritizing head, pelvic and trunk stability while accommodating hip and knee extensor spasms as required. By 24 months of age, a commercial wheelchair with dynamic seat and back was introduced, with custom seating and added padding to prevent injury from uncontrolled arm movements.

Power mobility, supported standing and supported stepping were all introduced by 12 months adjusted age and made available for use in the home environment. Parents were supported in using these devices to allow daily practice within natural routines, promoting opportunities for the children to be active, to initiate and to engage in problem solving. The early recognition that the twins were likely to function as GMFCS V and the ready access to assistive devices for extended trial, loan or purchase were essential for this outcome.

Recent research from Spain [14] and New Zealand [13] suggests that many clinicians do not use evidence-based tools such as the GMA-MOS or HINE to detect CP or to assist with early identification of whether a child is likely to be non-ambulant. Without early identification of this risk, many therapists wait too long, losing out on critical periods of neuro-plasticity, instead of providing assistive devices ON Time for children who are unable to maintain position and move independently. Families should be coached to provide affordances and routines-based, family-delivered strategies on a daily basis, rather than non-evidence-based resource-intensive handling techniques. Therapist engagement and partnership with families may be an important influence on successful outcomes [61].

In clinical practice, standers are often introduced from around 13 months (although 9–12 months is recommended) [27]; however, age-appropriate introduction of independent mobility devices is more variable. A survey of PTs in the United Kingdom suggests that many therapists only consider supported-stepping devices at older ages, once it is evident that children are not going to step any other way [62]. In contrast, an older survey of PTs in the United States (US) suggests earlier provision to promote development of walking, even for those who can later walk without the support [63]. A survey of paediatric PTs and OTs in Canada and the US showed that although 80% reported positive views towards the early use of power mobility, few provided or supported this in practice [64]. These differences illustrate the gap between therapist knowledge and the integration of evidence into practice.

ON-Time use of equipment improved **functioning** for the girls described in this case report, and eating, socializing and communicating were the most important goals initially. Equipment afforded more opportunities for the girls and increased participation in family meals, celebrations and outings. This illustrates the conclusion that, despite low-level experimental evidence, adaptive seating improves activity and participation within the family contextual environment for children with non-ambulant CP [65].

Beginning mobility experiences were **fun** for the girls, allowing them to experience independent movement like other children their age and to engage in play. Expert consensus supports the use of power mobility for children who will never walk to promote independence, overall development, self-initiated behaviour and learning, regardless of whether they will become functional, independent power wheelchair users in all environments [29]. A recent systematic review supports the positive impact of early childhood use of switch-adapted ride-on toys on children's activity and participation, and on family life [66]. A large qualitative study found that supportive mobility devices including power,

manual and supported stepping devices were equated with participation, independence and freedom [67]. In a similar manner to the family in this case report, parents of children classified at GMFCS V perceived power mobility experiences between one and three years of age as positively influencing their child's sense of autonomy and participation [68].

The 24-h activity guidelines for children with CP recommend at least 60 min of moderate activity daily and strenuous activity for at least 20 min 2–3 times weekly [69]. However, research suggests that only 2–7% of children at GMFCS I-III engage in even moderate activity [70]. Children at all GMFCS levels need to reduce sedentary behaviour and even increasing light activity is better than no activity [69]. For children at GMFCS V, **fitness** relates primarily to changes of position (such as standing or being supported upright in a stepping device), supported mobility or movement opportunities (e.g., stepping, hippotherapy, swimming, adapted dancing, frame-running, etc.), reducing sedentary behaviour (time spent in tilted or reclined positions during the day) and postural management to reduce time spent in static, asymmetrical and harmful positions. Postural management interventions positively impact hip health for children with non-ambulant CP [54], and the parents' quotes illustrate their belief in the importance of postural management for maintaining the girls' physical health and the impact this has on their **future**.

Supported stepping positions children eye-to-eye with peers and has a positive influence on self-esteem, confidence, communication and participation with others, which may allow them to make **friends** [28]. For young children, participation with family and siblings is more common, but at school age, interaction with other children becomes very important. The family quotes highlight the influence of devices such as standers, power wheelchairs and eye gaze technology in increasing the girls' opportunities for friends.

**Family** life is impacted by how the assistive devices assist or challenge daily life. The twins' older sister pointed out how adaptive equipment increased the girls' opportunities, and parents commented that the 'pros outweighed the cons', and they 'couldn't fathom life without it'. In other studies, parents have reported that positioning and mobility devices increase the child's function and lighten caregiving, but environmental barriers may limit use [71]. Parents of children who require several different assistive devices have been reported to have difficulties with lack of space and home access, particularly for larger devices such as power wheelchairs [72]. The family in this study was very proactive in seeking funding for home accessibility modifications and an adapted vehicle. They also successfully advocated in their community for provision of accessible playground equipment and recreational facility access, thus increasing opportunities for their own children and others. These factors may have influenced the more positive comments noted in this report.

*Limitations*

The retrospective and descriptive nature of this report limits the strength of its conclusions; however, the strength of a case report lies in the depth of description that allows comparison with similar cases, potentially widening transferability. Feasibility of ON-Time (age-appropriate) introduction of power mobility, supported standing and stepping devices, in addition to adaptive seating, was described and illustrated using valid and reliable tools. Although some measures and classifications were completed retrospectively (from medical charts and videos), ratings and scores were agreed by a team of researchers with varying experience with the cases and measures, potentially increasing dependability. Lived-experience data reflect both memories of past device use and application or influence of these experiences on current abilities and outcomes and may be considered a form of member checking, as well as a means of increasing credibility and confirmability.

**5. Conclusions**

This report illustrates that introducing adaptive seating, standing, stepping and power mobility devices is feasible within the first 2 years of life, provided that motor prognosis is identified early; the family is supported in identifying meaningful goals and opportunities

for device use; parents/caregivers are trained in how to use assistive devices to enhance child participation and engagement in desired family activities and routines; and appropriate assistive devices are available for loan or purchase in a timely manner. For the twins in this case report, the ON-Time introduction and use of adaptive seating, standing, stepping and power mobility interventions resulted in increased opportunities for fun, functioning, fitness, enhanced family participation, increased opportunities for friends and promoted future opportunities and overall health.

Children under 24 months functioning at GMFCS V are often held by adults or placed in lying positions for most of their awake time. The environment must be enriched and modified to afford development of cognition, vision, language, function and self-advocacy. When children are not required to request and deny interactions and cannot explore their environment using their own self-generated movements, they do not learn about spatial awareness, cause–effect or object permanence. Without assistive devices to afford upright positioning and mobility in order to enhance participation, these children are left behind, and overall development is negatively affected.

Tools such as GMA-MOS and HINE should be regularly used in clinical practice to allow early identification of children at the highest risk of being GMFCS IV or V. Appropriate assistive devices may take time to procure, and early initiation of the process is essential if children at GMFCS IV or V are to be provided with age-appropriate, ON-Time experiences of standing, stepping and self-initiated mobility. Families and caregivers require support and formal training in order to effectively integrate interventions and assistive devices into daily life and routines and positively address the F-words for child development: functioning, family, fitness, fun, friends and future. Assistive device interventions are evidence-based and should be considered a child's human right and standard-of-care.

**Author Contributions:** Conceptualization, methodology, data review and analysis, R.W.L., A.J.C. and G.S.P.; semi-structured interview conduct and transcription, A.J.C.; writing—original draft preparation, R.W.L.; table and figure preparation, R.W.L., G.S.P. and A.J.C.; writing—review and editing, A.J.C. and G.S.P. All authors have read and agreed to the published version of the manuscript.

**Funding:** This research received no external funding.

**Institutional Review Board Statement:** Ethical review and approval were waived by University of British Columbia, Children's and Women's Health Centre of British Columbia Ethics Review Board for this study and publication due to its descriptive nature.

**Informed Consent Statement:** Written informed consent was obtained from the parents and assent from the children to publish this paper. Informed consent was obtained from all participants in this study.

**Data Availability Statement:** All data included in this study can be found in the published manuscript.

**Acknowledgments:** The authors would like to thank the children and family described in this case report for generously sharing their time, family photos and videos for the purpose of this publication. The authors would also like to acknowledge Peter Rosenbaum, Rachel Teplicky and CanChild for permission to personalize and adapt the ICF-F-word graphic.

**Conflicts of Interest:** R.W.L. and A.J.C declare no conflict of interest. G.S.P has worked as an educational consultant for Prime Engineering, a manufacturer of standing and stepping devices. This relationship had no influence on this unfunded study.

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
