# Peer review of "Power Mobility, Supported Standing and Stepping Device Use in the First Two Years of Life: A Case Report of Twins Functioning at GMFCS V"

_disabilities, doi:10.3390/disabilities3040032_

Round 1
Reviewer 1 Report
Comments and Suggestions for Authors
Dear Authors,
Thank you for interesting reading. You show a deep knowledge in your field of practice. And I do believe a detailed case study can support the understanding of the importance of beginning standing and mobility interventions early - as it impacts later development.
However, I suggest some minor amendments to improve structure and clarity.
The impact of early intervention on later development needs to be mentioned in the background - as it relates to Figure 6 in the results. The two parts of the aim has to be more apparent in the result section (3) and in the beginning of the discussion (4), maybe also in the conclusion, please see commentaries in the attached PDF-file. Even though the text is already long I also find you need to add some text to explain some content or clarify concepts that readers may not be familiar with.
As a whole the detailed descriptions in the manuscript add to understanding the benefits of providing children with severe disabilities opportunities for early standing and mobility - both for children and family.

Author Response
Thank-you for your helpful review. Please find attached detailed responses to your comments.

Reviewer 2 Report
Comments and Suggestions for Authors
The introduction is well written and thorough; however, the transition to the F-words is somewhat abrupt.
It is unclear how old the twins are at this time – it is documented that they provided assent. How was their assent obtained. Later in the document it refers to the use of augmented and assistive communication – was this employed to obtain their assent? It appears that they are eleven, so this is nine years post?
The methodology is unclear - it is described as a case report but videos were reviewed and interviews recorded which is a form of data collection - it would be helpful to describe it as a case report that includes quantitative and qualitative data.
Were there no differences at all between the two girls? There tonal and postural patterns were identical. It would help to discuss them separately prior to page 7. How did the positioning support in the stander differ between the two girls? In addition – the timeline narrative is lengthy and would benefit from a timeline with key milestones and trials indicated? The introduction of the tables sooner would be helpful to the reader.
There is an extensive list of instruments – it would be helpful to the reader to create tables.
A description of a typical day and routines-based interventions would be helpful to appreciate the impact of the assistive technology on the children’s mobility and ability to interact with their environment.
The photos demonstrate positioning but do not provide full body images to appreciate postural changes and
The reporting of the interviews is cumbersome – a review of common themes shared by both parents and perhaps differences would be providing more insight than reading what appears to be an abridged transcript.
If the children are 11 – It would be helpful to have a thorough clinical presentation including cited outcome measures in addition to the F word graphic.
The discussion includes a paragraph on physical activity which seems out of place as physical activity was not a primary focus of early mobility in this case report.
The limitations refer to methodology such as coding of videos which was not described in the methods section.
Conclusions advocate for use of outcome measures, yet this case report is about mobility and use of adaptive equipment and assistive technology.
Overall, the content of the paper is confusing - providing more tables or figures to support the text would be helpful. Articles have been published on benefit from early mobility - however, very few report longitudinal data or long-term implications. If this is 9 years after the girls received this type of mobility supports, then a discussion of the long-term implications and the girls' current function would add greatly to the strength of the article.
Author Response
Thank-you for your helpful review. Please see attached file for detailed responses to your comments and questions.

Round 2
Reviewer 1 Report
Comments and Suggestions for Authors
Dear authors,
Thank you for the revision, which increased understanding of the F-words and how they relate to the retrospetive description of early mobility in two cases.
By clarifying the F-words also those unfamiliar with this way to express what's in the ICF may become interested in reflecting childrens intervention outcomes in the light of this family-friendly terminology.
Most important is the clarification of how early intervention influences on outcomes later in development - which emphasizes the the importance for the childrens future.
Thank you for an important contribution supporting early intervention
Reviewer 2 Report
Comments and Suggestions for Authors
Thank you for your revisions - the revised manuscript provides greater clarity on the study as a whole and the methodology. Readers will benefit from the longitudinal perspective of this paper and are afforded the opportunity to reflect on the role of positioning and mobility.